# Development of Robust Tablet Formulations with Enhanced Drug Dissolution Profiles from Centrifugally-Spun Micro-Fibrous Solid Dispersions of Itraconazole, a BCS Class II Drug

**DOI:** 10.3390/pharmaceutics15030802

**Published:** 2023-03-01

**Authors:** Stefania Marano, Manish Ghimire, Shahrzad Missaghi, Ali Rajabi-Siahboomi, Duncan Q. M. Craig, Susan A. Barker

**Affiliations:** 1School of Pharmacy, University College London (UCL), 29-39 Brunswick Square, London WC1N 1AX, UK; 2Colorcon Inc., Global Headquarters, 275 Ruth Road, Harleysville, PA 19438, USA

**Keywords:** centrifugal spinning, microfibre, amorphous solid dispersion, sucrose, poorly water-soluble drug, itraconazole, dissolution enhancement, supersaturation, tabletting, oral formulation

## Abstract

Fibre-based oral drug delivery systems are an attractive approach to addressing low drug solubility, although clear strategies for incorporating such systems into viable dosage forms have not yet been demonstrated. The present study extends our previous work on drug-loaded sucrose microfibres produced by centrifugal melt spinning to examine systems with high drug loading and investigates their incorporation into realistic tablet formulations. Itraconazole, a model BCS Class II hydrophobic drug, was incorporated into sucrose microfibres at 10, 20, 30, and 50% *w*/*w*. Microfibres were exposed to high relative humidity conditions (25 °C/75% RH) for 30 days to deliberately induce sucrose recrystallisation and collapse of the fibrous structure into powdery particles. The collapsed particles were successfully processed into pharmaceutically acceptable tablets using a dry mixing and direct compression approach. The dissolution advantage of the fresh microfibres was maintained and even enhanced after humidity treatment for drug loadings up to 30% *w*/*w* and, importantly, retained after compression into tablets. Variations in excipient content and compression force allowed manipulation of the disintegration rate and drug content of the tablets. This then permitted control of the rate of supersaturation generation, allowing the optimisation of the formulation in terms of its dissolution profile. In conclusion, the microfibre-tablet approach has been shown to be a viable method for formulating poorly soluble BCS Class II drugs with improved dissolution performance.

## 1. Introduction

Appropriate formulation of Biopharmaceutics Classification System (BCS) Class II drugs, characterised by low solubility and high permeability, is essential to maximising their oral bioavailability and thereby minimising waste. Many formulation approaches have been investigated in an attempt to increase the rate and extent of dissolution of BCS Class II drugs, with solid dispersion technology being especially well studied. Over the last decade, advanced fabrication technologies such as electrospinning and centrifugal spinning (melt and solution) have been used to prepare solid dispersions in the form of polymer- or sugar-based nano- and micro-fibres [1,2,3,4,5,6]. Typically, the formulation design is for the drug to be molecularly dispersed within the fibres, leading to enhanced in vitro dissolution and in vivo bioavailability. Several excellent reviews on this area of pharmaceutical formulation science have recently been published, providing an in-depth analysis of the current understanding of the subject [7,8,9,10,11].

Despite the potential dissolution and oral bioavailability advantages of fibre-based formulations, they require further processing into a patient-friendly format before use. However, due to the particular morphological features of nano- or micro-fibres (high surface area to volume ratio, high porosity, and low density), the incorporation of these systems into a conventional oral dosage form capable of being manufactured reproducibly on a large scale may be challenging. In particular, content (dose) uniformity in the final product is critical and will be dependent on adequate mixing of the fibres and the additional formulation excipients, potentially requiring a milling stage with concomitant risks of both chemical and physical instability of the drug. Tabletting (compression) of “fluffy” material such as fibre mats to obtain a robust product may be mechanically difficult, carries a risk of inducing recrystallisation of an amorphous material, and may result in a decrease in the dissolution rate compared with the native uncompressed fibres due to effective surface area changes. Finally, if the dissolution and oral bioavailability advantages of the original fibres are dependent on the drug being present in the amorphous state, the risks of recrystallisation on storage, leading to deleterious changes in the dissolution profile over time, need to be evaluated and mitigated by judicious choice of packaging and storage conditions.

There have been relatively few studies on the downstream processing of electrospun fibres. Electrospun drug-loaded fibre mats have been manually folded and placed in gelatine capsule shells [12], although this was more of a processing aid for dissolution studies than an attempt to develop a viable pharmaceutical product. Some studies [13,14,15,16] have shown that electrospun drug-loaded fibre mats can be manually compressed after direct loading of the fibre mat into the tablet die. Other authors have ground the fibre mats to effect size reduction and mixed them with large quantities of conventional tabletting excipients prior to compression, reflecting a more conventional oral tablet formulation [17,18,19,20,21,22]. Two recent studies have focussed on the mixing step in the conversion of electrospun drug-loaded fibre mats into tablets. Fülöp et al. [23] showed that it is possible to prepare low-dose tablets with acceptable content uniformity from electrospun drug-loaded fibre mats after high-shear dry mixing with conventional tabletting excipients on a reasonable development lab scale (500 g) and compression on a rotary tablet press. Szabó et al. [24] successfully applied continuous manufacturing processing techniques (mixing, compression) with inline analysis to prepare tablets from electrospun drug-loaded microfibre mats; in this case, the milled fibres were treated as a starting material. A detailed study on the mechanical and compression behaviour of electrospun drug-loaded nanofibre mats has recently been published [25]. The presence of the drug, at 10% *w*/*w* loading, was shown to affect the mechanical properties of the nanofibre mat, in that the drug-loaded samples exhibited greater stiffness and lower ductility than the placebos. Interestingly, only the drug-loaded nanofibrous tablets showed clear fracture behaviour during diametral compression testing after preparation, with the placebo nanofibrous tablets and the tablets prepared from equivalent physical mixtures all showing continuous yielding and tablet shape change instead. The study by Démuth et al. [26] illustrated the need for judicious choice of excipients in developing tablet formulations from amorphous solid dispersions. They studied electrospun nanofibres containing itraconazole, a poorly water-soluble drug. A total drug release of only 80% of the theoretical amount was observed from tablets containing the water-insoluble lubricant magnesium stearate, which was attributed to the amorphous itraconazole in the nanofibres crystallising on the magnesium stearate during the dissolution test. When a more water-soluble lubricant (sodium stearyl fumarate) was used instead, dissolution was both faster and more extensive, with drug release values of >95% of the theoretical value being achieved.

Much less attention has been paid to the technique of centrifugal spinning (sometimes described as rotary spinning), either in the solution mode or the melt mode, as a means of preparing micro- and nanofibres that can be further processed for pharmaceutical use. Two studies [27,28] used centrifugal solution spinning of aqueous solutions of drug and polymer to generate drug-loaded nanofibre mats, which were then micronised, mixed with standard tabletting excipients, and compressed into tablets. Centrifugal melt spinning was used in two studies [29,30] to prepare nanofibre mats of drug and sucrose/polyvinylpyrrolidone (PVP), which were then directly compressed into tablets with no further processing. In both cases, dissolution from the tablets was slightly slower than from the nanofibre mats, and in one case [29], the tablets were observed to be sticky, indicating that, although the nanofibre mats could be compacted, further work is required to convert the nanofibres into a useable formulation.

In our previous work [3,4], we have used centrifugal melt spinning to successfully manufacture sucrose-based microfibres containing three clinically significant BCS Class II drugs (itraconazole, olanzapine, and piroxicam) at 10% *w*/*w* drug loading. Upon manufacture, both the drug and the carrier (sucrose) were observed to exist in an amorphous state in the microfibres, although small amounts of crystalline drug were detected in the piroxicam-loaded microfibres. Under both sink and non-sink conditions, all the fresh drug-loaded microfibres showed significantly superior dissolution behaviour to the raw drugs and to simple physical mixtures of equivalent formulation, with supersaturation being observed in the non-sink testing protocol. The amorphous nature of both the drug and the microfibre matrix, the solubilizing capability of sucrose, and the high surface area to volume ratio of the microfibres all contributed to these effects. Surprisingly, all three drug-loaded microfibres retained their dissolution and supersaturation advantages after exposure to 75% relative humidity (RH) at 25 °C for short periods (around 1 day), which resulted in the crystallisation of the sucrose carrier leading to the collapse of the microfibres to powder. No change in the dissolution profile was observed after 8 months of further exposure to 25 °C/75% RH. Interestingly, while olanzapine and piroxicam recrystallised along with the sucrose, itraconazole appeared to be present in an amorphous state in the humidity-treated samples. These results show that there is considerable potential for the use of centrifugally melt-spun microfibres in enhancing the dissolution profiles of poorly water-soluble drugs. Additionally, these studies indicate that the relationship between the physical form of the system and the drug dissolution behaviour may be more complex than anticipated, with recrystallisation on exposure to high humidity conditions not necessarily leading to the decrease in dissolution rate that may be commonly expected.

This then raises the question of whether it is possible to deliberately recrystallise freshly prepared drug-loaded microfibres by exposure to an environment with controlled high humidity, causing the collapse of the microfibres into powder, and use this powder as the drug source for further processing into conventional tablets for ease of administration to patients. Such an approach should have the dual advantages of retaining the dissolution benefit of the microfibres while using a source material (powder) that is more easily processed into tablets than the freshly prepared microfibres themselves. It is this question that is explored in the current study. Here we have investigated the feasibility of preparing tablet formulations using standard pharmaceutical excipients and processing methods that contain drug-loaded microfibres produced by centrifugal melt spinning and exposed to high humidity conditions. Our aim is to develop a tablet product with a fast dissolution profile that is both scalable and economically viable. Itraconazole was chosen as the model drug for this study, as there is a particular need for an improved oral formulation of itraconazole, given its relatively low oral bioavailability (55%), and the variability in plasma levels reported following oral administration under different feeding regimes [31,32]. However, as a low level of drug loading in the microfibres (10% *w*/*w* in our previous studies) will inevitably lead to a low drug content in the final tablets, the effects of increasing the itraconazole content of the microfibres have also been investigated. The flow properties of the drug-loaded microfibres, alone and in formulation mixtures, were studied, along with their mixing and segregation potentials. Selected formulations were compressed, and their pharmaceutical performance was assessed in terms of compliance with standard mandatory pharmacopoeial tests and their disintegration and dissolution profiles.

## 2. Materials and Methods

### 2.1. Materials

Itraconazole (ITZ) (molecular weight 705.64 g/mol, melting point 166 °C, glass transition temperature 60 °C) was purchased from Watson Noke Scientific Ltd. (Suzhou, China), and sucrose was obtained from Sigma-Aldrich Co. (St Louis, MO, USA). All buffer salts used for the dissolution media, as well as acetic acid (≥99.85%), acetonitrile (≥99.93%), and sodium n-dodecyl sulphate (≥99%), were purchased from Sigma-Aldrich (Taufkirchen, Germany). Avicel PH102^®^ was purchased from FMC (Cork, Ireland). StarTab^®^ was obtained from Colorcon (Harleysville, PA, USA). Compressol SM^®^ (SPI-Pharma, Wilmington, DE, USA), Kollidon CL-F^®^ (BASF, Ludwigshafen, Germany), and Compritol 888 ATO^®^ (Gattefossé UK Ltd., Ascot, UK) were obtained as gift samples. All other chemical reagents were of analytical grade.

### 2.2. Methods

#### 2.2.1. Preparation of Fresh Microfibres by Centrifugal Melt Spinning

Sucrose-based itraconazole-loaded microfibres with varying itraconazole content (20, 30, and 50% *w*/*w*) and pure drug microfibres (100% *w*/*w*) were prepared using the previously described centrifugal melt spinning process [3,4]. For samples containing up to 50% *w*/*w* drug loading, microfibres were prepared by spinning the appropriate physical mixture of sucrose and itraconazole at a fixed rotational speed of 2400 rpm and operating temperature of 197 °C. Pure crystalline itraconazole was directly spun in the absence of sucrose at a fixed rotational speed of 2400 rpm and a range of operating temperatures to produce 100% *w*/*w* pure itraconazole microfibres.

For comparison purposes, equivalent quench-cooled solid dispersion samples were prepared in situ in a DSC pan by heating sucrose-itraconazole physical mixes and pure itraconazole at a rate of 10 °C/minute to a final temperature 3 °C above the melting temperature of the mix (previously determined), holding isothermally for 1 minute, then cooling at a rate of 20 °C/minute to −20 °C. These samples were analysed immediately using the same modulated temperature differential scanning calorimetry (MTDSC) protocol as the microfibre samples [3,4].

#### 2.2.2. High-Humidity Treatment of Fresh Microfibres

All freshly prepared samples were stored at 25 °C/75% RH in open glass vials, as previously described (4), to induce recrystallisation and microfibre collapse. The samples were stored in a sealed desiccator containing a saturated salt solution of sodium chloride to generate the 75% RH condition. The desiccator was stored in a 25 ᵒC chamber to maintain the temperature.

#### 2.2.3. Physical and Chemical Characterisation of Fresh and Aged Microfibres

The methods used here to characterise the drug-loaded sucrose microfibres have been described in detail in our previous work [3,4], so only brief details are given here.

Drug content of the microfibres was assessed immediately after preparation and after 30 days exposure to 25 °C/75% RH. Itraconazole was extracted using 50:50 acetonitrile:pH 6.8 phosphate buffer, then measured using reversed-phase HPLC (Synergi 4 μm Polar-RP 80 Å, 50 × 3 mm column (Phenomenex, Macclesfield, UK)) with a mobile phase of 50:50 acetontrile: water-acetic acid (0.1% *v*/*v*) and UV detection at 264 nm.

The physical state of itraconazole and sucrose within the microfibre samples was measured immediately after preparation and then monitored on a daily basis by MTDSC and X-ray Powder Diffraction (XRPD). For the MTDSC studies, a fully calibrated Q2000 (TA Instruments Q2000, New Castle, DE, USA) with a refrigerated cooling system and a dry nitrogen sample purge was used. All samples were tested in PerkinElmer 40 μL aluminium pans with pinholed lids, with an underlying heating rate of 2 °C/minute and a ±0.212 °C modulation amplitude over a 60 s period. For the XRPD studies, a MiniFlex diffractometer (RigaKu, Tokyo, Japan) was used. XRPD patterns were recorded using diffraction angles (2θ) from 5° to 50° (step size 0.05°; time per step 0.2 s).

Scanning Electron Microscopy (SEM) was used to assess the morphology of microfibre samples on preparation and after humidity treatment. Samples were gold-coated (20 nm) under vacuum using a Quorum Q150T Turbo-Pumped Sputter Coater (Quorum Technologies, Laughton, UK) and then imaged with a Quanta 200F instrument (FEI, Hillsborough, OR, USA).

Non-sink dissolution testing was performed on the fresh and aged microfibre samples, each sample containing the equivalent of 10 mg of drug. The dissolution medium was 50 mL of phosphate buffer (pH: 6.8) containing 0.1% *w*/*v* of sodium dodecyl sulfate (SDS), maintained at 37 ± 0.2 °C in a shaking incubator. One (1) mL samples were withdrawn at pre-determined time intervals and filtered through a 0.22 μm Millipore Millex^®^ GT filter. The drawn volume was replaced with the same amount of blank dissolution medium at 37 ± 0.2 °C. Drug concentration was measured using the HPLC-UV system after appropriate dilution.

#### 2.2.4. Preparation of Powder Blends

Powder blends of aged microfibres and relevant excipients were prepared by initial mixing by geometric dilution with a mortar and pestle. Powder blends were then transferred into plastic containers and further mixed for 10 min using a Turbula T2G blender (Willy A. Bachofen AG, Muttenz, Switzerland). The powder blends were analysed for their flow properties and segregation potential, as described below.

#### 2.2.5. Powder Flow Analysis

Powder flow was assessed via analysis of bulk and tapped densities, according to the method described in the United States Pharmacopeia (USP) Chapter <616>, “Bulk Density and Tapped Density of Powders.” A Sotax TD2 tap density tester (Hopkinton, MA, USA) was used with a 100 mL glass measuring cylinder and 25 g of sample. Initial ($V_{0}$) and final ($V_{F}$) volumes were measured after 1 tap and 1250 taps, respectively. The Carr index was calculated as in Equation (1) below.
(1)$$Carr\;index\;\left(\%\right)=100\times \left(V_{0}-V_{F}\right)/V_{0}$$

#### 2.2.6. Powder Segregation Analysis

A novel approach to segregation testing, based on the bulk and tapped density analytical method, was used here to assess the segregation tendency of microfibres in excipient mixtures. The Sotax TD2 tap density tester (Hopkinton, MA, USA) was used, but the conventional single measuring cylinder was replaced with a 100 mL outer plastic cylinder with a removable base and a set of seven stacking inner plastic cylinders to enable sampling at specific heights in the powder bed. Powder mixes (25 g) were tapped 100 times, and then samples weighing 300 mg (corresponding to the target tablet weight used in later studies) were taken at the evenly-spaced sampling points.

Drug content in the segregation test samples was measured by dissolving samples containing a theoretical load of 3 mg itraconazole in 100 mL of methanol, followed by appropriate dilution for UV detection at 260 nm. Solutions were sonicated for at least 30 min to ensure complete drug dissolution prior to the analysis. No interference from the excipients or methanol was observed at the detection wavelength.

#### 2.2.7. Preparation of Tablets

The formulation of the initial eleven batches of tablets is shown in Table 1. For each batch, 25 g of powder mix was prepared. The powder blends were first prepared as described above in the absence of lubricant, then the lubricant was added and the mixture blended for another minute. Tablets with a target weight of 300 mg were prepared by compression using an instrumented eccentric tablet press (Atlas Auto T8, Specac, Kent, UK) equipped with 10 mm round, flat-faced punches. Tablets were produced at compression forces of 10, 16, 20, and 26 kN. Further batches of tablets based on the analysis of these batches were produced in an analogous fashion.

#### 2.2.8. Physical and Analytical Characterisation of Tablets

Tablet dimensions were measured using a digital caliper (Manchester, UK). Tablet crushing strength was measured using an 8M hardness tester (Thun, Switzerland) on 10 randomly selected tablets according to USP chapter <1217> “Tablet Breaking Force.” Subsequently, the tablet tensile strength T (MPa) was calculated as in Equation (2) below, where F (N) is the tablet crushing strength, d (mm) is the tablet diameter, and h (mm) is the tablet thickness (Fell and Newton, 1970).
(2)T=2F/πdh

Tablet friability was measured according to USP chapter <1216> “Tablet Friability Test.” A unit of ten pre-weighed tablets was rotated at 25 rpm for 4 min using a TAR 20 Friability tester (Erweka, Germany). Tablets were then dusted off and reweighed, and the percentage weight loss was calculated.

The disintegration times of six random tablets from each batch were measured at 37 ± 2 °C in 900 mL of distilled water on a ZT54 dissolution tester (Erweka, Milford, CT, USA) according to USP Chapter <701> “Disintegration.”

Selected batches of tablets were assessed for drug content uniformity as per the requirements of USP Chapter <905> “Uniformity of Dosage Units.” Ten tablets of each batch were crushed individually, and itraconazole extracted and analysed as described above for the segregation test samples.

Non-sink dissolution testing was performed on selected tablets in the same manner as for the microfibres. However, in this case, the tablets were tested intact, and the itraconazole content varied from 3 mg to 45 mg.

Solid-State ^13^C NMR Spectroscopy was used to assess the physical state of the itraconazole in the tablets, using the procedures described in our previous work [4]. High-resolution spectra were recorded using cross-polarisation (CP), MAS, high-power proton decoupling, and total suppression of sidebands (TOSS). The tablets were crushed prior to testing in order to facilitate the experiment.

#### 2.2.9. Statistical Analysis

All results are expressed as mean ± SD. For the dissolution studies, the maximum drug concentration in solution (C_max_) and the time of its occurrence (T_max_) were obtained from the drug concentration–time profiles. The supersaturation profiles between formulations were compared by measuring the area under the curve (AUC). Data from different formulations were compared for statistical significance by one-way analysis of variance (ANOVA). Differences were considered statistically significant at *p* < 0.05.

## 3. Results and Discussion

### 3.1. Preparation of Microfibres with Increasing Drug Loading and Characterisation of the Fresh Microfibres

Microfibres were successfully prepared by centrifugal melt spinning of physical mixtures of sucrose and itraconazole as described above. The calculated percentage yields (% of theoretical, mean ± SD, n = 6) were 94.6 ± 1.5, 93.4 ± 1.7, and 94.1 ± 1.4, respectively, for the 20, 30, and 50% *w*/*w* drug-loaded microfibres, similar to the value of 95.4 ± 2.1 observed for the 10% *w*/*w* sample [4]. Drug content uniformity values (% of theoretical, mean ± SD, n = 6) of 98.6 ± 1.5 and 99.4 ± 1.3 for the 20 and 30% *w*/*w* samples, respectively, were again similar to that seen for the 10% *w*/*w* sample, 99.5 ± 1.1 [4], indicating full incorporation and homogeneous distribution of the drug in the microfibre product. However, at 50% *w*/*w* itraconazole loading, a higher mean value and greater variation were observed, i.e., 117.1 ± 9.8, suggesting that the drug may not be homogeneously distributed within the sucrose matrix at high drug incorporation, possibly due to the limited loading capacity of the sucrose carrier. This is discussed in more detail below. Surprisingly, itraconazole was able to form microfibres alone, with no sucrose carrier. The optimum temperature for spinning pure itraconazole was found to be 183 °C, with a yield of 65.7 ± 4.3% of theoretical. At lower temperatures, no microfibres were formed even though the drug was molten, whereas at higher temperatures, the yields were lower as some of the drug stuck to the spinneret. The lower spinning temperature for pure itraconazole compared with the mixed itraconazole-sucrose systems is a consequence of itraconazole’s lower melting point of 166 °C [33] compared with that of sucrose (186 °C) [34]. We believe this is the first report of pure itraconazole microfibres being formed using any production technique.

As shown in Figure 1, under SEM, all freshly prepared itraconazole-loaded sucrose microfibres showed smooth surface morphology with no defects or evidence of surface or bulk drug crystallisation.

The mean microfibre diameter was independent of itraconazole content, with values of circa 7 µm being observed for all samples, as detailed in Table 2. However, visual inspection of the 50% *w*/*w* itraconazole-loaded samples and closer analysis of their SEM images showed the presence of thin (1 to 5 µm diameter), grey-coloured microfibres with a visibly different texture from the bulk of the microfibres. These thinner microfibres were similar in appearance and diameter to the pure itraconazole microfibres (3.22 ± 2.54 µm, mean ± SD), suggesting that some of the thinner microfibres in the 50% *w*/*w* itraconazole-loaded sample were in fact pure itraconazole microfibres.

XRPD diffractograms of all fresh drug-loaded sucrose microfibres (20, 30, and 50% *w*/*w*) and the pure drug microfibres showed the typical broad halo pattern of an amorphous material, as shown in Figure 2, as had previously been observed for the freshly prepared drug-free and 10% *w*/*w* drug-loaded sucrose systems.

Figure 3 shows the MTDSC reversing heat flow traces of the spun microfibres and the equivalent quench-cooled samples. 

All samples showed glass transitional behaviour, indicating the generation of amorphous dispersions. Interestingly, the quench-cooled sucrose-itraconazole samples all showed two separate glass transitions assigned to the individual components, with that of sucrose occurring at 68 ± 4.3 °C, indicated with a red arrow, and that of itraconazole occurring at 60 ± 0.2 °C, indicated with a blue arrow. These data indicate phase separation, i.e., lack of miscibility, of the two components, as discussed previously [35] for other systems. Further evidence of phase separation is offered by the presence of the two endothermic events occurring at around 74 and 90 °C, ascribed to the formation of a chiral nematic mesophase of pure itraconazole (90 °C) and rotational restriction of the molecules (74 °C) upon cooling from the melt, which are reversible upon re-heating, leading to the observed endotherms [36]. Both of these endothermic transitions and the glass transition are clearly visible for the quench-cooled pure itraconazole sample here. Microfibres containing 20 and 30% *w*/*w* itraconazole showed a single mixed-phase glass transition, indicated with a green arrow, at 74.7 ± 1.1 °C and 73.2 ± 1.3 °C, respectively, similar to that observed for the 10% *w*/*w* drug-loaded sample at 74.1 ± 1.9 °C [4]. Conversely, the thermal behaviour of the 50% *w*/*w* itraconazole-loaded microfibres was more similar to that of the corresponding quench-cooled sample, with both individual substance glass transitions and both itraconazole endothermic events being observed. It is interesting to note that the endothermic transitions are less pronounced for the microfibres, possibly indicating a lower degree of phase separation compared with the equivalent quench-cooled sample. The pure itraconazole microfibres showed the expected glass transition and endothermic transitions.

Taken together, these initial characterisation data suggest that the melt centrifugal process may increase the degree of mixing and miscibility of the drug (itraconazole) in the carrier (sucrose) compared with the simple melt quenching method, possibly due to the application of high centrifugal forces, with the true limit of miscibility being between 30 and 50% *w*/*w* of itraconazole. In the 50% *w*/*w* itraconazole-loaded samples, at least some of the excess drug appears to be ejected as pure drug microfibres rather than solidifying as conventional drug particles. This observation also provides an explanation for the more variable content uniformity data seen for the 50% *w*/*w*-loaded samples than those with lower drug content: each sample taken for analysis is likely to contain a different proportion of the various populations of microfibres, leading to greater variability in the results.

### 3.2. High Humidity Treatment of Fresh Microfibres

All high-drug-loaded microfibres showed sucrose recrystallisation after storage at 25 °C/75% RH in open containers, as demonstrated by the appearance of sucrose Bragg peaks in the XRPD diffractograms and collapse of the microfibre structure, as described previously for the 10% *w*/*w* system [4]. Interestingly, the time required for the microfibres to collapse increases as a function of the itraconazole content. The 10% *w*/*w* sample was observed to collapse within 24 h, whereas microfibres containing higher amounts of itraconazole required significantly longer times to do so. Specifically, systems containing 20, 30, and 50% *w*/*w* itraconazole were seen to collapse in 4.2 ± 1.3, 7.4 ± 1.9, and 19.7 ± 2.4 days, respectively (n = 3 for each formulation). This is likely due to differences in the water uptake tendency of the microfibres with the higher concentrations of this highly lipophilic drug. Our previous studies [4] on centrifugally spun sucrose microfibres containing 10% *w*/*w* of olanzapine (log P = 2.2), piroxicam (log P = 3.06), or itraconazole (log P = 5.66) showed that the more hydrophobic the drug, the lower the moisture uptake as determined by dynamic vapour sorption, and the slower the sucrose recrystallisation and microfibre collapse. This was attributed to the increased hydrophobicity of the microfibre surface in the presence of the drug. It is logical, therefore, to expect that increasing the quantity of itraconazole in the microfibres would further retard the diffusion of water through the hydrophilic sucrose matrix, slowing down the sucrose recrystallisation and the collapse of fibrous structure.

For all subsequent studies, all microfibre formulations were held at 25 °C/75% RH in open containers for 30 days in order to ensure that the sucrose had fully recrystallised in all samples and comparisons between formulations were valid.

### 3.3. Characterisation of the 30-Day Humidity-Treated Microfibres

After exposure of freshly prepared samples to high humidity conditions, microfibres with 20 and 30% *w*/*w* itraconazole collapsed into powders with similar average particle size and morphology to those observed for the previously investigated system containing 10% *w*/*w* itraconazole [4]. At 50% *w*/*w* drug loading, there are more particles with a more elongated morphology compared with all other systems. However, in all cases, the moisture-treated itraconazole-sucrose systems collapsed into significantly smaller particles than the drug-free sucrose sample. Figure 1 shows the SEM micrographs of the fresh microfibres with their corresponding diameter frequency diagrams and the SEM images of the collapsed powder after humidity treatment. Table 2 summarises the size data for the samples studied here and the 10% *w*/*w* drug-loaded and pure sucrose samples for comparison. The pure itraconazole microfibres showed no change in appearance after exposure to the high humidity conditions for 30 days, with the measured diameter of 3.13 ± 2.50 µm (mean ± SD) showing no significant difference to that of the fresh samples (*p* > 0.05). The different collapse behaviour of the microfibres (total collapse for the pure sucrose and 10, 20, and 30% *w*/*w* itraconazole-loaded microfibres; partial collapse of the 50% *w*/*w* itraconazole-loaded microfibres; no observable collapse for the pure itraconazole microfibres) suggest that the dominant factor in the collapse process is the sucrose. For the 50% *w*/*w* itraconazole-loaded samples, those microfibres containing sucrose will collapse after humidity treatment, whereas microfibres containing pure or almost pure drug will not, leading to the observation of a mixed population of particles and microfibres after treatment.

Figure 4 shows the XRPD diffractograms for the high humidity-treated microfibre formulations.

The presence of the characteristic Bragg peaks for crystalline sucrose and the absence of peaks corresponding to crystalline itraconazole confirmed that the sucrose had largely or fully recrystallised from all the aged sucrose-itraconazole systems, whereas it is inferred that the itraconazole remained in the amorphous state. We had previously observed this behaviour for the itraconazole-sucrose system containing 10% *w*/*w* drug [4]. However, no recrystallisation of the pure itraconazole microfibre system was observed after 30 days, as indicated by the presence of a broad halo pattern and the absence of any Bragg peaks for crystalline itraconazole. The MTDSC reversing heat flow traces of the high humidity-treated microfibre formulations are shown in Figure 5.

The absence of the glass transition associated with sucrose (circa 68 °C) and the presence of the glass transition corresponding to itraconazole (circa 60 °C), followed by the two endothermic transitions associated with amorphous itraconazole (circa 74 °C and 90 °C) confirmed that sucrose fully recrystallised from all the itraconazole-sucrose systems, whereas itraconazole remained in the amorphous state.

Drug content uniformity was measured for the moisture-treated samples, with values (% of theoretical, mean ± SD, n = 6) of 99.1 ± 1.2, 97.9 ± 1.7 and 106.3 ± 11.2 for the 20, 30, and 50% *w*/*w* samples, respectively. These are essentially unchanged from the fresh samples, suggesting that the microfibre collapse did not adversely affect the drug distribution in the samples.

Overall, physical and chemical analysis of the humidity-treated microfibres shows that the collapse is due to recrystallisation of the sucrose carrier and that the drug remains in the amorphous form, possibly explained by its high hydrophobicity, which reduces its interaction with water during storage.

### 3.4. Non-Sink Dissolution Testing of Fresh and Aged Microfibres

In our previous study [4], humidity-treated 10% *w*/*w* itraconazole-loaded sucrose microfibres exhibited an unexpected increase (circa 1.25-fold) in the solubility of itraconazole, observed under non-sink conditions, compared with the freshly prepared microfibres, which themselves showed very significant itraconazole solubility increases (circa 8-fold) compared with the pure crystalline drug or an equivalent physical mix. The solubility advantage of the treated samples was maintained even after 8 months of further exposure to high humidity conditions. Here, we have studied the dissolution and solubility behaviour of the samples with higher drug content, both fresh and aged for 30 days at 25 °C/75% RH, under the same non-sink test conditions and with the same total amount of drug in each experiment. The concentration–time profiles of the fresh and aged samples are shown in Figure 6 and Figure 7, respectively.

Examining the fresh samples first, in all cases, the initial drug dissolution process was relatively rapid, forming supersaturated solutions with itraconazole concentrations far exceeding the measured crystalline itraconazole equilibrium solubility (approximately 7 µg/mL). It is interesting to note the behaviour of the pure amorphous itraconazole microfibres, as this gives an indication of the amorphous itraconazole equilibrium solubility (approximately 40 µg/mL) as well as allowing interrogation of the role of sucrose in the dissolution and supersaturation processes.

Both the rate of supersaturation generation and the peak concentration of the itraconazole in solution are affected by the drug-carrier ratio. The time required (T_max_) to reach the maximum level of drug supersaturation (C_max_) increases with increasing drug loading, with T_max_ values being approximately 1, 2, and 3 h for the 20, 30, and 50% *w*/*w* itraconazole-loaded microfibres, respectively. Samples containing 20 and 30% *w*/*w* of itraconazole reached an equivalent maximum level of supersaturation (*p* > 0.05) (C_max_ = 57.7 ± 5.9 and 59.2 ± 4.7 µg/mL, respectively), following which drug precipitation occurred, with the final observed dissolved concentrations stabilising at the level of the equilibrium solubility of amorphous itraconazole. In contrast, at 50% *w*/*w* drug loading, the C_max_ was significantly lower (*p* < 0.05), reaching only 44.9 ± 3.9 µg/mL and rapidly decreasing down to the pure amorphous itraconazole solubility. The 10% *w*/*w* drug-loaded sample showed the same trend as the 20 and 30% *w*/*w* drug-loaded samples: T_max_ was shorter at approximately 30 min, and C_max_ was not significantly different (*p* > 0.05) at 56.78 ± 5.9 µg/mL [4].

This observed pattern of supersaturation behaviour can be explained by a combination of the effect of the dissolving hydrophilic sucrose and the solid-state characteristics of the microfibres. At 50% *w*/*w* drug loading, the sucrose-itraconazole system is at least partially phase-separated, and the actual amount of sucrose in the dissolving sample is low. It is reasonable to suggest that the rapid dissolution of this small amount of sucrose is able to affect only marginally the initial drug dissolution step, leaving the phase-separated amorphous drug to regulate the final dissolution step, hence the final drug concentration cannot substantially exceed the solubility of the equivalent pure amorphous drug. At higher sucrose ratios (lower drug content), where the drug and carrier are molecularly dispersed in the solid microfibres, the initial drug dissolution profiles are more controlled by the concomitant dissolution of the carrier and some of the amorphous drug. This will lead to a more consistent C_max_, as this will ultimately be dependent on the extent of the interactions between the two components and the maximum increase in solubility of itraconazole that may be effected by the presence of sucrose. The shorter T_max_ with the lower drug contents is a reflection of the higher sucrose:drug ratio, leading to greater interaction between the two components initially and a faster rate of supersaturation generation. The decrease in measured drug concentration from the C_max_ down to the level of equilibrium amorphous drug solubility is likely caused by the complete dissolution of the carrier, leaving the remaining undissolved amorphous drug to control the rest of the process. Even though the very high levels of supersaturation are maintained for a relatively short period (up to about 2 h), this may be sufficient to allow enhanced oral absorption, as considered in more detail below.

Examining now the dissolution–supersaturation profiles for the humidity-treated samples, at first sight, the profiles of the aged samples are very similar to those of the fresh samples. As expected based on the physical analysis described above, there was no change in the behaviour of the pure amorphous itraconazole microfibres, and the dissolution profile of the 50% *w*/*w* itraconazole sample has decreased to match that of the pure itraconazole microfibres, losing the initial solubility benefit seen in the fresh sample. In contrast, the initial dissolution rate of the 20 and 30% *w*/*w* drug-loaded samples decreased compared with the fresh samples, but the T_max_ values remained the same and the supersaturation (C_max_) levels were increased, to 68.15 ± 4.12 and 69.17 ± 7.45 µg/mL, respectively, a similar profile to the 10% *w*/*w*-loaded samples previously described [4]. The C_max_ values for the aged 20 and 30% *w*/*w* drug-loaded samples were not significantly different from each other but were significantly different (*p* < 0.05) from the corresponding fresh samples. This may be explained by the slower dissolution of crystalline sucrose in the aged samples compared with the amorphous sucrose in the fresh samples, leading to slower initial dissolution of the drug but preventing a too-rapid buildup of supersaturation, which would then lead to rapid precipitation back to the equilibrium amorphous itraconazole solubility levels. These observations are in agreement with the theoretical model proposed by Han and Lee [37], in which rapid supersaturation generation above a critical value is followed by rapid desaturation based on the decreasing energy barrier to nucleation and precipitation seen at high supersaturation levels.

The overall dissolution performance can be further evaluated by comparing the area under the curve (AUC) of the dissolution–supersaturation profiles for the same quantity of drug, displayed in Figure 8, for the full 24 h of study. The 10, 20, and 30% *w*/*w* itraconazole-loaded aged samples all showed a statistically significant increase in AUC compared with their fresh counterparts, although the effect was much greater for the 10% *w*/*w* drug-loaded sample. Importantly, the aged 10% *w*/*w* drug-loaded samples showed a statistically significant better AUC performance than all other aged samples. Overall, the dissolution advantage of the microfibre generation and humidity treatment seems to be greater for the lower drug loading samples, but this must be balanced against the downstream formulation constraints and product size issues, as will be discussed in the next section.

### 3.5. Tablet Development—Powder Characterisation

Tablet development focussed on the 10 and 30% *w*/*w* drug-loaded aged microfibres, as these respectively showed the greatest dissolution/supersaturation advantage and the likely benefits of size reduction of the final product due to greater drug incorporation. Wet granulation was not considered to be a desirable method of tablet production here due to the risk of damage to the microfibres because of the presence of water (or other granulation solvents) and heat; a direct compression method was therefore selected. However, this approach requires careful consideration of the mixing and flow properties of the individual components and the formulation as a whole; hence, the collapsed microfibres were studied without further processing.

Both the aged microfibre samples showed poor flow characteristics, as would be expected given their morphologies, with Carr indices of > 30%. Binary mixtures of the aged microfibre samples with Compressol SM^®^ or StarTab^®^ showed significant improvements in flow behaviour: even at 60% *w*/*w* microfibre content, the Carr indices were < 20%, indicating good to fair flow. Conversely, binary mixtures with Avicel PH102^®^ showed no improvement in flow properties compared with the microfibre samples alone, except at the lowest microfibre incorporation (10% *w*/*w*). The segregation potential of the binary mixtures was assessed using a novel segregation cell based on a tapped density apparatus. Here, the arrangement of an outer plastic cylinder with a removable base and a set of stacking inner plastic cylinders, rather than the normal single measuring cylinder, allows for ease of sampling at specific heights in the powder bed, after tapping to simulate the vibration and particle movement expected in the tabletting processes.

The mixing and segregation profiles of the aged microfibres were assessed using the 10% *w*/*w* itraconazole-loaded system. No segregation was seen with binary mixtures based on StarTab^®^ containing 10 to 60%*w*/*w* microfibres, with drug content values of all individual unit-sized samples (300 mg total weight) comfortably within the range of 95 to 105% of theoretical, significantly better than the pharmacopoeial content uniformity specification of 85 to 115% of theoretical. Binary mixtures based on Avicel PH102^®^ or Compressol SM^®^ showed inappropriate segregation behaviour at low microfibre loadings (10% *w*/*w*), with content uniformity data outside the pharmacopoeial limits, but this improved as the microfibre loadings increased, such that at 60% *w*/*w* microfibre loading, the content uniformity values were within the range of 95 to 105% of theoretical. The lack of segregation seen with StarTab^®^ is attributable to its morphology, with the high specific surface area and porosity providing greater opportunity for mechanical interlocking of the microfibres and the carrier particles. Subsequent experiments showed that a pre-mixing step of microfibres, equivalent to 10% *w*/*w* loading in the final mix, with StarTab^®^ prior to dilution with either Avicel PH102^®^, Compressol SM^®^, or a 1:1 mix of these two excipients, led to non-segregating powders, with all measured drug content values being in the range of 95 to 105% of theoretical. These results suggest that, as long as the processed microfibres are initially mixed with StarTab^®^, other components with alternative functionalities may be added to generate a fully functional tablet formulation.

### 3.6. Tablet Development—Physical Characterisation

Using the aged 10% *w*/*w* itraconazole-loaded microfibres, nine different tablet formulations, labelled F1 to F9 and shown in Table 1, were produced via a direct compression process, i.e., pre-mixing of the microfibres with StarTab^®^ to prevent segregation, dry mixing with other ingredients, lubrication, and compression. In all formulations, the level of the disintegrant (crospovidone (Kollidon CL-F^®^)) was kept constant at 5% *w*/*w*, and the level of the lubricant (glyceryl dibehenate (Compritol 888 ATO^®^)) was maintained at 1.5% *w*/*w*. The formulations are divided into three groups (Groups 1, 2, and 3) based on the content of the drug-loaded microfibres (10, 30, and 50% *w*/*w*, respectively). Within each group, the amount of StarTab^®^ relative to Avicel PH102^®^ and Compressol SM^®^ was increased in the ratios of 1:1:1 (formulations F1, F4, and F7), 2:1:1 (formulations F2, F5, and F8), and 3:1:1 (formulations F3, F6, and F9). Finally, for comparative purposes, Group 4 included the highest possible loading of the microfibres (93.5% *w*/*w*) with just the disintegrant and lubricant in formulation F10 and the raw, unprocessed ingredients in the same ratios in formulation F11. All formulations were compressed at four compression forces: 10, 16, 20, and 26 kN.

Figure 9A–D shows the tensile strength–compression force and disintegration time–compression force curves for tablets from all 11 formulations.

Both F10 and F11 mixtures stuck to the tablet punches during compression, illustrating the adhesive nature of their major constituents. F11 tablets showed low tensile strengths (circa 0.6 MPa) and correspondingly short disintegration times (circa 3.5 min) at all compression forces, along with lamination during the tensile strength testing. In contrast, F10 tablets showed an increase in tensile strength as a function of compression force up to 20 kN (ranging from 1.9 to 2.9 MPa), then a decrease at 26 kN, along with lamination during testing at this compression force. No significant differences (*p* > 0.05) in the disintegration time were observed for F10 tablets, with all tablets disintegrating in 20 to 23 min. All F11 tablet batches and the F10 tablet batch showing lamination failed the pharmacopoeial friability test; all other F10 tablet batches passed this test.

Within Groups 1, 2, and 3, a roughly linear increase in tablet tensile strength with increasing compression force was observed, with a concomitant increase in disintegration time. A closer inspection of the data highlights that both the percentage of aged microfibres and the ratio of the excipients affected the behaviour of the tablets. Within each group, tablets with the lowest relative content of StarTab^®^ (F1, F4, F7) generally showed numerically greater values of tensile strength than the formulations with the middle (F2, F5, F8) and highest (F3, F6, F9) relative content of this excipient, although the differences were statistically significant (*p* < 0.05) only for Group 1 formulations at all compression forces and Group 2 formulations at the highest compression forces. A similar pattern was observed with the disintegration times. Comparing between groups, increasing the aged microfibre content from 10% *w*/*w* to 30% *w*/*w* resulted in an increase in tensile strength and disintegration time for formulations compressed at the same force and with the same excipient ratio. For example, F2 (Group 1) tablets compressed at 16 kN showed mean values of 2.1 MPa and 30 s, respectively, whereas the mean values for F5 (Group 2) tablets were significantly higher (*p* < 0.05) at 2.6 MPa and 6.2 min. However, as the microfibre content increases still further to 50% *w*/*w*, the differences become less significant. For example, F4 (Group 2) tablets compressed at 20 kN show statistically (*p* > 0.05) similar mean tensile strength and disintegration time values (2.9 MPa and 8.2 min, respectively) compared with the equivalent F7 (Group 3) tablets (2.8 MPa and 10.2 min, respectively). No sticking or lamination was observed for any of these formulations (F1 to F9), and all batches passed the pharmacopoeial friability test.

These results suggest that, although it is possible to directly compress the aged microfibres into tablets, a more considered formulation approach is required to produce robust tablets. A complex relationship exists between the formulation components in terms of the effect on the tensile strength and dissolution time of the resultant tablets, but the dominant formulation factor appears to be the content of the aged microfibres, with the excipient ratio playing a smaller role, most obviously observed at lower microfibre contents when the excipient content is correspondingly greater. The effect of increasing the compression force is predictable in that increasing compression force leads to increased tensile strength and disintegration time. However, these findings also demonstrate that it is possible to fine-tune the tensile strength and disintegration time of the tablets by varying the microfibre content, excipient ratios, and compression forces, potentially allowing the development of tablets for different purposes.

### 3.7. Tablet Development—Non-Sink Dissolution Testing

Formulations F3 and F9 were chosen for the non-sink dissolution study, representing the extremes of microfibre content, tensile strength, and disintegration time and having the same excipient ratio. Replicate formulations, labelled as F3* and F9*, containing the aged 30% *w*/*w* itraconazole-loaded microfibres were also studied. This sample selection allows investigation of the effects of both drug content and rate of supersaturation generation on the overall dissolution performance of the tablets. Table 3 shows the characterisation data of all these batches.

Drug content uniformity and weight uniformity data were excellent, with all tablet batches clearly passing the relevant pharmacopoeial specifications. No significant differences (*p* > 0.05) were observed in any test between equivalent batches varying only in the drug content, i.e., F3/F3* and F9/F9*. This is a significant result, as it demonstrates that the microfibres show the same mechanical and formulation behaviour irrespective of drug content in the range of 10 to 30% *w*/*w*.

Concentration–time profiles, generated under the same non-sink conditions as used earlier for the microfibres, of the tablets and the corresponding uncompressed blends and pure crystalline itraconazole are shown in Figure 10 (F3 and F3*) and Figure 11 (F9 and F9*), respectively.

Examining the fast-disintegrating F3 formulations first, the uncompressed microfibres and the tablets compressed at 10, 16, and 20 kN showed broadly equivalent dissolution performance, achieving and maintaining supersaturation (C_max_) at levels of approximately 20 µg/mL, i.e., approximately 2.8-fold higher than the pure crystalline drug solubility, although there was a trend towards slightly lower rates of drug dissolution from the tablets as the compression force increased. Tablets compressed at the highest compression force (26 kN) did not show supersaturation, and the corresponding dissolution profiles were similar to those of the pure crystalline drug. F3* microfibres and tablets showed similar dissolution behaviour to the equivalent F3 microfibres and tablets in relation to the effects of compression force and supersaturation generation. However, for F3* tablets compressed at forces up to and including 20 kN, some drug precipitation occurred shortly after the initial high values of C_max_ (in the range of 58 to 68 µg/mL) were attained, leading to the maintenance supersaturation levels being approximately 50 µg/mL, i.e., approximately 7.1-fold higher than the pure crystalline drug solubility and approximately 25% higher than the solubility of amorphous itraconazole established earlier. There was a trend toward lower peak C_max_ levels and lower maintenance supersaturation levels with increasing compression force. The plateau supersaturation levels obtained for the F3* tablets, which contain 9 mg itraconazole, were approximately 2.5-fold higher than those for the F3* tablets, which contain 3 mg itraconazole, demonstrating that the dissolution advantage of the microfibres is preserved at the higher drug loading, even if there is not a completely linear relationship between drug loading and plateau drug concentrations.

Both F9 and F9* tablets compressed at 26 kN force showed dissolution profiles indistinguishable from the raw crystalline drug. F9 uncompressed microfibres and tablets compressed at 10 kN both showed an initial fast dissolution and high C_max_ (approximately 85 and 83 µg/mL, respectively), with subsequent precipitation and stabilisation of supersaturation levels at approximately 50 and 56 µg/mL, respectively. The higher drug-loaded F9* formulation showed similar behaviour. In this case, the C_max_ values were approximately 83 and 95 µg/mL, respectively, for the uncompressed microfibres and tablets compressed at 10 kN, with the supersaturation concentration plateauing for both products at approximately 48 µg/mL. Closer inspection of the data reveals that, for both F9 and F9*, the tablets compressed at 10 kN showed significantly (*p* < 0.05) lower dissolution at the very early stages of the experiment, up to 15 min for F9 and 30 min for F9*, than the corresponding powder blend, which may be attributable to the 5 min disintegration time for these tablets reducing the initial rate of dissolution. This slower buildup of supersaturation is then responsible for the retention of the higher supersaturation concentrations in the tablets compared with the powder blends, following the model of Han and Lee [37].

Taken together, these results indicate that the rate and extent of supersaturation can be controlled by varying the drug content of the formulation and the disintegration time of the tablets, which will then affect the total amount of drug available for absorption in the intestine. The overall effect of this may be quantified by calculating the 24 h AUC values of the dissolution–supersaturation profiles: values (mean ± sd) of 480 ± 14, 1223 ± 18, 1448 ± 44 and 1178 ± 36 µg/mL.h were observed for F3, F3*, F9, and F9* tablets compressed at 10 kN, respectively, with the equivalent value for crystalline itraconazole being 158 ± 4 µg/mL.h. The 3-fold increase in overall drug content between F3 and F3*, achieved via an increase in the drug loading of the microfibres (10 and 30% *w*/*w*), led to a 2.5-fold increase in both final sustained concentration and AUC. However, a similarly derived 3-fold itraconazole content increase between F9 and F9* did not result in a similar dissolution advantage, and indeed, a lower AUC was observed for the F9* tablets, indicating that, above a certain limit, increasing the drug content has no effect on the final dissolution profile. The 5-fold increase in overall drug content between F3 and F9, achieved via an increase in the microfibre content of the tablets (10 and 50% *w*/*w*), led to an almost 3-fold increase in both final sustained concentration and AUC. Conversely, the similarly derived 5-fold itraconazole content increase between F3* and F9* tablets resulted in a slight decrease in overall AUC, again suggesting that there is a maximal effect of increasing the content of the drug. This dose effect is attributable to the initial supersaturated concentrations exceeding the critical value whereby the energy barrier to recrystallisation is sufficiently low to allow precipitation to occur. From these data, F9 would appear to be the most beneficial formulation. However, the absorption site for itraconazole is the small intestine, which has a commonly accepted transit time of 3 to 5 h. Therefore, considering only the first five hours of the dissolution experiment to simulate the likely absorption window, the corresponding AUC (mean ± SD) values for F3, F3*, F9, F9* tablets, and crystalline drug are 103 ± 5, 272 ± 9, 361 ± 14, 307 ± 12, and 18 ± 2 µg/mL∙h, confirming that F9 is the best of the current formulations.

### 3.8. Tablet Development—Physical State of the Drug in the Compressed Tablets

The non-sink dissolution study highlighted that all tablet batches compressed at 26 kN showed dissolution profiles indistinguishable from the raw crystalline itraconazole, suggesting that compression at this force had resulted in the recrystallisation of the amorphous itraconazole in the microfibres. This was confirmed by ^13^C CP/MAS SSNMR analysis. Figure 12 shows the responses of F3 and F9* tablets, with the lowest (1% *w*/*w*) and highest (15% *w*/*w*) itraconazole content, respectively, compressed at 10 and 26 kN, and the peaks attributable to crystalline and amorphous drug [4] shown by arrows. All tablets compressed at 10, 16, and 20 kN showed the amorphous response, and all tablets compressed at 26 kN showed the crystalline response, indicating that the recrystallisation process is compression force dependent, with the critical force being between 20 and 26 kN. These results highlight the need to understand the effect of process variables on the physical structure and behaviour of the formulation and, by extension, the likely effect on the biological performance of the drug.

## 4. General Discussion and Future Perspectives

Building on our previous work [3,4], this study has shown that it is possible to prepare fully amorphous, one-phase itraconazole-loaded sucrose microfibres using melt centrifugal spinning, up to a drug content of 30% *w*/*w*. At 50% *w*/*w* drug loading, phase separation was seen, including the generation of pure itraconazole microfibres in the amorphous state. Fresh microfibres with up to 30%*w*/*w* itraconazole showed similar non-sink dissolution behaviour to that previously observed for the 10% *w*/*w* drug-loaded samples, i.e., a fast initial dissolution with high supersaturated concentrations being generated and subsequent partial drug precipitation with lower supersaturation concentrations being maintained for up to 24 h. Exposure for 30 days to 25 °C/75% RH resulted in the recrystallisation of the sucrose component of the microfibres, but the itraconazole remained in the amorphous state. The dissolution–supersaturation advantage seen in the fresh samples was maintained and even enhanced in the aged systems, ascribed to the slower initial dissolution of crystalline sucrose preventing a too-rapid rise in supersaturation and hence reducing the extent of precipitation thereafter.

An in-depth tablet formulation development study was carried out using aged itraconazole-loaded microfibres and commonly used direct compression tabletting excipients. To enhance mixing and prevent segregation, a pre-mixing step of the microfibres with StarTab^®^, a partially pregelatinised starch, was necessary prior to subsequent mixing with the remainder of the excipients, lubrication, and compression. The content of the aged microfibres was observed to have the greatest influence on the physical behaviour of the tablets, with the excipient mix playing a lesser role. Non-sink dissolution studies showed that the dissolution–supersaturation advantage of the itraconazole-loaded (10 and 30% *w*/*w*) aged microfibres was maintained in the tablet formulations containing 10% *w*/*w* microfibres and even enhanced for a microfibre content of 50% *w*/*w* in the final tablets when tablets were compressed at low and medium forces. However, compression at the highest force resulted in the recrystallisation of the amorphous drug, and the dissolution profiles became indistinguishable from those of the crystalline raw drug. All tablet batches tested for dissolution passed all relevant pharmacopoeial tests, with excellent drug content uniformity and weight uniformity being demonstrated. Tablet formulation F9, containing 50% *w*/*w* microfibres composed of 10% *w*/*w* itraconazole, showed the greatest AUC advantage when measured over 5 h to simulate intestinal transit and itraconazole’s likely absorption window, or 24 h to simulate transit through the entire gastro-intestinal tract. This was attributed to a relatively slow rate of supersaturation generation and a subsequent slow precipitation rate, which led to a high maintenance supersaturation level. The lack of further increase in C_max_ or AUC seen with the microfibres and tablets with the highest drug loading (F9* compared with F9) demonstrates the overall limit of supersaturation that may be achieved with this approach and this drug. Additionally, it allows a rough estimation of the critical supersaturation concentration, above which drug precipitation occurs rapidly, as described by Han and Lee [37]. This value can be estimated here as being in the range of 90 to 100 µg/mL.

F9 and F9* tablets, which contain 50% *w*/*w* microfibres, showed longer disintegration times (around 5 min) compared with their equivalent formulations containing 10% *w*/*w* microfibres (a few seconds), which was thought to contribute to the slower onset of supersaturation. Extending this approach and developing a tablet with an even longer disintegration time may then lead to greater overall dissolution performance. In this case, the rate of supersaturation generation and hence the potential for drug precipitation would be expected to be reduced further while still maintaining the overall dissolution advantage of the amorphous state of the drug and the increase in solubility due to the presence of sucrose in the microfibres. This may be achieved by reducing the content of disintegrant in the tablet formulation. Increasing the compression force, which would also be expected to increase disintegration time and slow the initial dissolution rate, carries the risk of causing the in situ recrystallisation of the amorphous itraconazole, thus losing all benefit of this formulation approach.

From a commercial perspective, the potential scalability and speed of the centrifugal melt spinning process are important considerations. On a lab scale, 10 g of powder mix can be converted into microfibres in approximately 5 min once the equipment is at the correct temperature. The spinning process has the potential to be made into a (semi)-continuous process, as long as there is a balance between feeding of the starting materials (the powder mix) and collection of the product (the microfibres); hence, it is potentially scalable into a commercially viable process. However, as there is a need to humidity-treat the microfibres prior to formulation into tablets, the overall process will need to remain a batch process.

## 5. Conclusions

This study has demonstrated that fully amorphous sucrose microfibres can be prepared with high drug loading by a centrifugal melt-spinning process. Under non-sink conditions, fresh microfibres showed a significant dissolution and supersaturation advantage compared with the raw drug and physical mixtures of the drug and sucrose, which was maintained after humidity treatment (25 °C/75% RH for 30 days), which caused recrystallisation of the sucrose and collapse of the microfibres. It is possible to directly compress the aged microfibres to form tablets, although sticking to the punches was observed. A detailed tablet formulation study using aged microfibres as the drug source demonstrated that high-quality tablets can be prepared using a direct compression approach; these tablets easily passed all relevant pharmacopoeial specifications. A pre-mixing step with StarTab^®^ was required to overcome the flow and segregation issues caused by the morphology of the collapsed microfibres. Importantly, the dissolution advantage of the microfibres was retained after compression into tablets. By varying the disintegration rate and drug content of the tablets, the rate of supersaturation generation and subsequent drug precipitation can be controlled, allowing the optimisation of the formulation in terms of dissolution profile. Overall, this investigation showed that the microfibre-tablet approach to formulating poorly soluble BCS Class II drugs leads to improved dissolution behaviour of the drug, which in turn should lead to enhanced oral bioavailability of the drug.

## Figures and Tables

**Figure 1 pharmaceutics-15-00802-f001:**
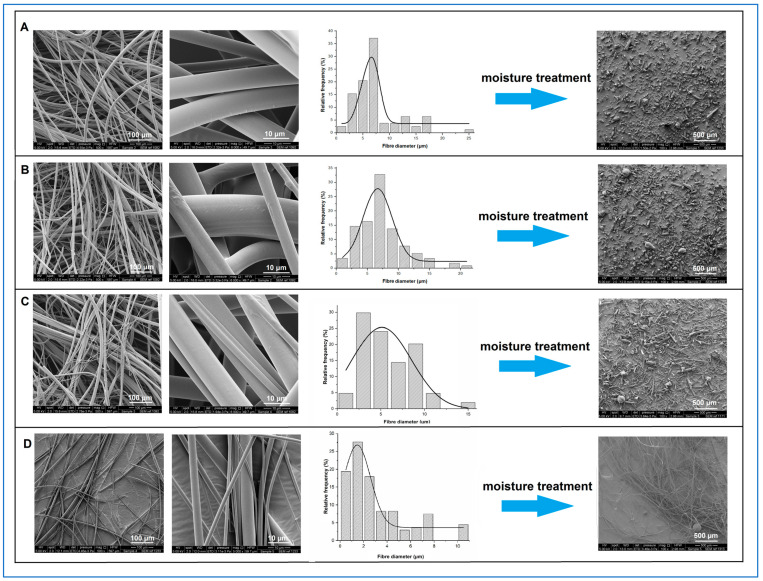
From left to right: SEM micrographs of freshly prepared itraconazole-loaded sucrose microfibres (500× and 6000× magnification), microfibre diameter frequency diagrams, SEM images of the corresponding samples (100× magnification) after 30 days exposure to 25 °C/75%RH. (**A**) 20% *w*/*w*, (**B**) 30% *w*/*w*, (**C**) 50% *w*/*w* itraconazole-loaded sucrose microfibres, and (**D**) 100% *w*/*w* itraconazole microfibres.

**Figure 2 pharmaceutics-15-00802-f002:**
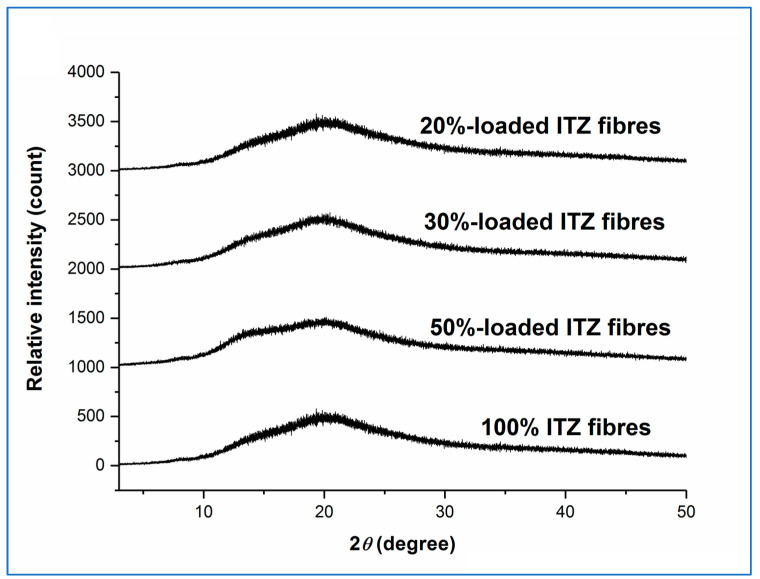
XRPD diffractograms of freshly prepared itraconazole-loaded sucrose microfibres (containing 20, 30, and 50% *w*/*w* itraconazole) and pure itraconazole microfibres. (ITZ = itraconazole).

**Figure 3 pharmaceutics-15-00802-f003:**
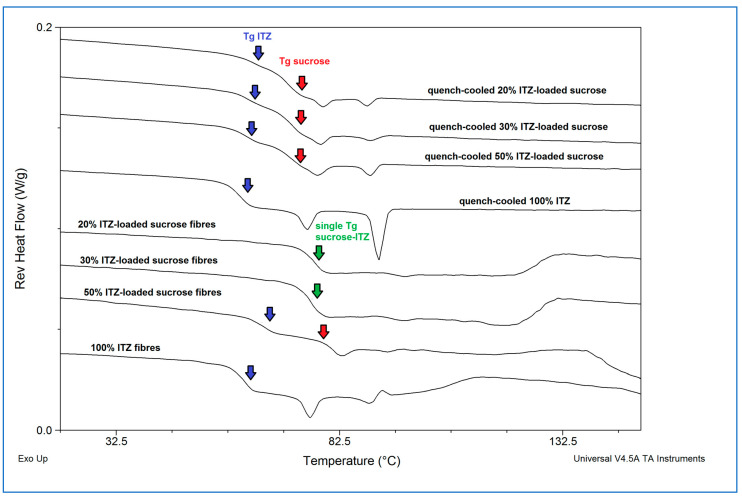
Reversing heat flow traces of freshly prepared itraconazole-loaded sucrose microfibres (containing 20, 30, and 50% *w*/*w* itraconazole), pure itraconazole microfibres, and solid dispersions with equivalent compositions prepared by quench-cooling from the melt. (ITZ = itraconazole).

**Figure 4 pharmaceutics-15-00802-f004:**
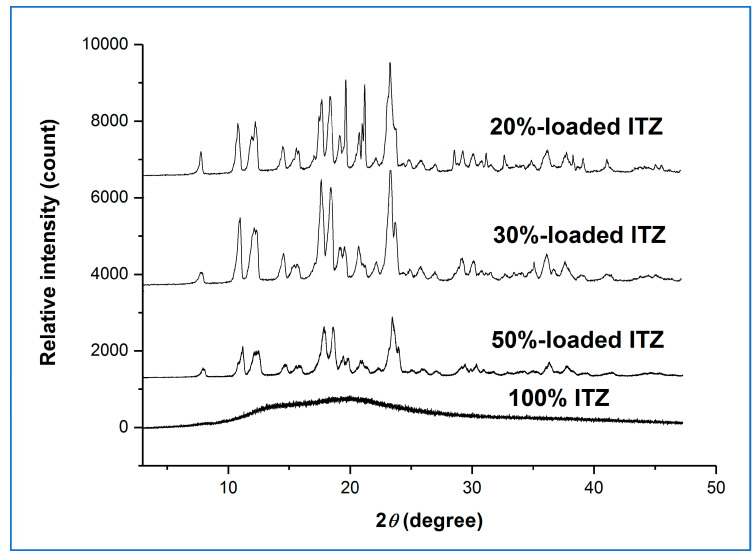
XRPD diffractograms of itraconazole-loaded sucrose microfibres (containing 20, 30, and 50% *w*/*w* itraconazole) and pure itraconazole microfibres after 30 days exposure to 25 °C/75% RH. (ITZ = itraconazole).

**Figure 5 pharmaceutics-15-00802-f005:**
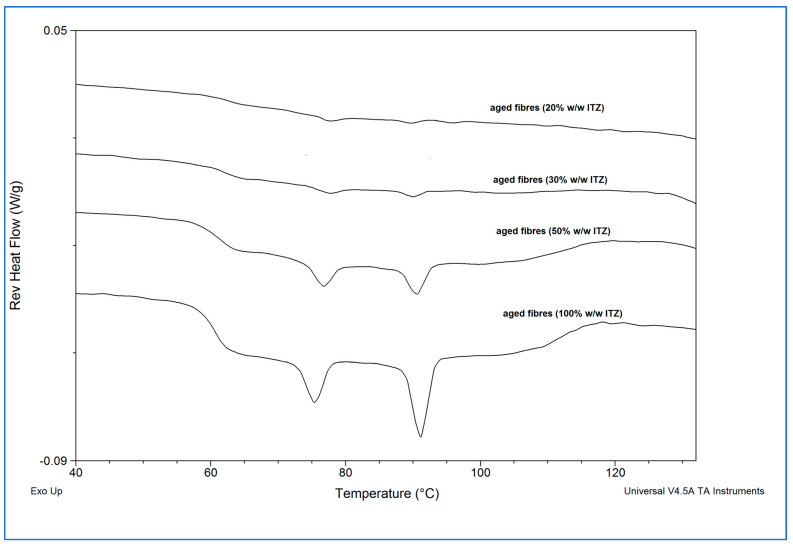
Reversing heat flow traces of itraconazole-loaded sucrose microfibres (containing 20, 30, and 50% *w*/*w* itraconazole) and pure itraconazole microfibres after 30 days exposure to 25 °C/75% RH. (ITZ = itraconazole).

**Figure 6 pharmaceutics-15-00802-f006:**
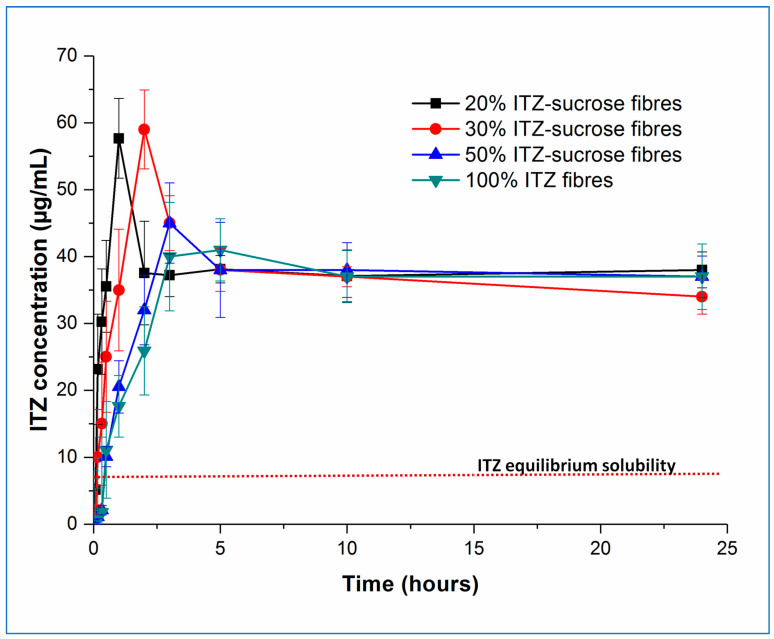
Dissolution–supersaturation profiles obtained under non-sink conditions for freshly prepared itraconazole-loaded sucrose microfibres (containing 20, 30, and 50% *w*/*w* itraconazole) and pure itraconazole microfibres. The red dotted line indicates the equilibrium solubility of crystalline itraconazole. (ITZ = itraconazole).

**Figure 7 pharmaceutics-15-00802-f007:**
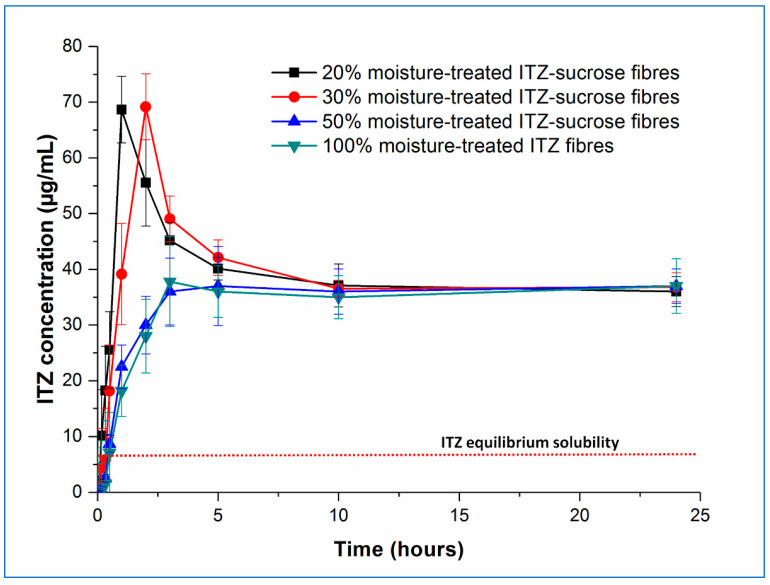
Dissolution–supersaturation profiles obtained under non-sink conditions for itraconazole-loaded sucrose microfibres (containing 20, 30, and 50% *w*/*w* itraconazole) and pure itraconazole microfibres after 30 days exposure to 25 °C/75% RH. The red dotted line indicates the equilibrium solubility of crystalline itraconazole. (ITZ = itraconazole).

**Figure 8 pharmaceutics-15-00802-f008:**
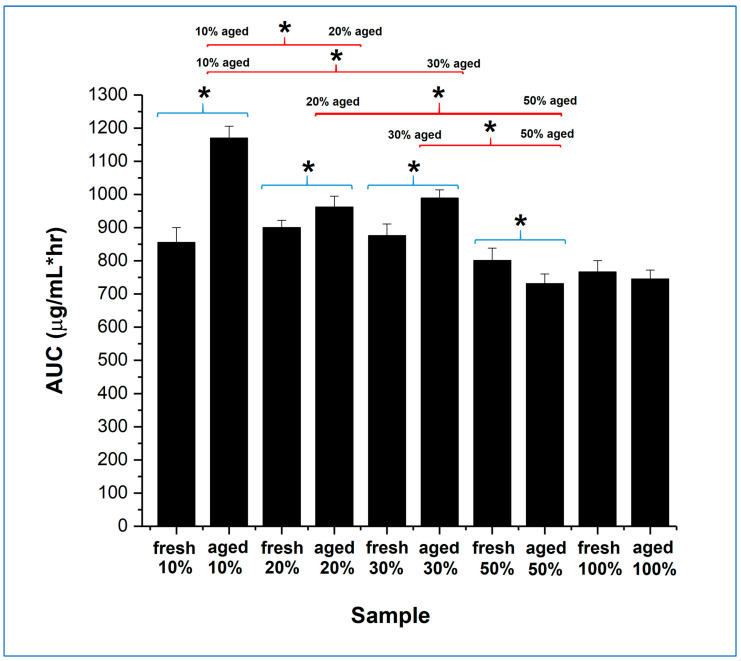
Overall (24 h) AUC of the dissolution–supersaturation profiles obtained under non-sink conditions for itraconazole-loaded sucrose microfibres (containing 20, 30, and 50% *w*/*w* itraconazole) and pure itraconazole microfibres, both freshly prepared and after 30 days exposure to 25 °C/75% RH (aged). * with blue brackets indicates a significant difference (*p* < 0.05) between fresh and aged samples of the same formulation. * with red brackets indicates a significant difference (*p* < 0.05) between different aged formulations.

**Figure 9 pharmaceutics-15-00802-f009:**
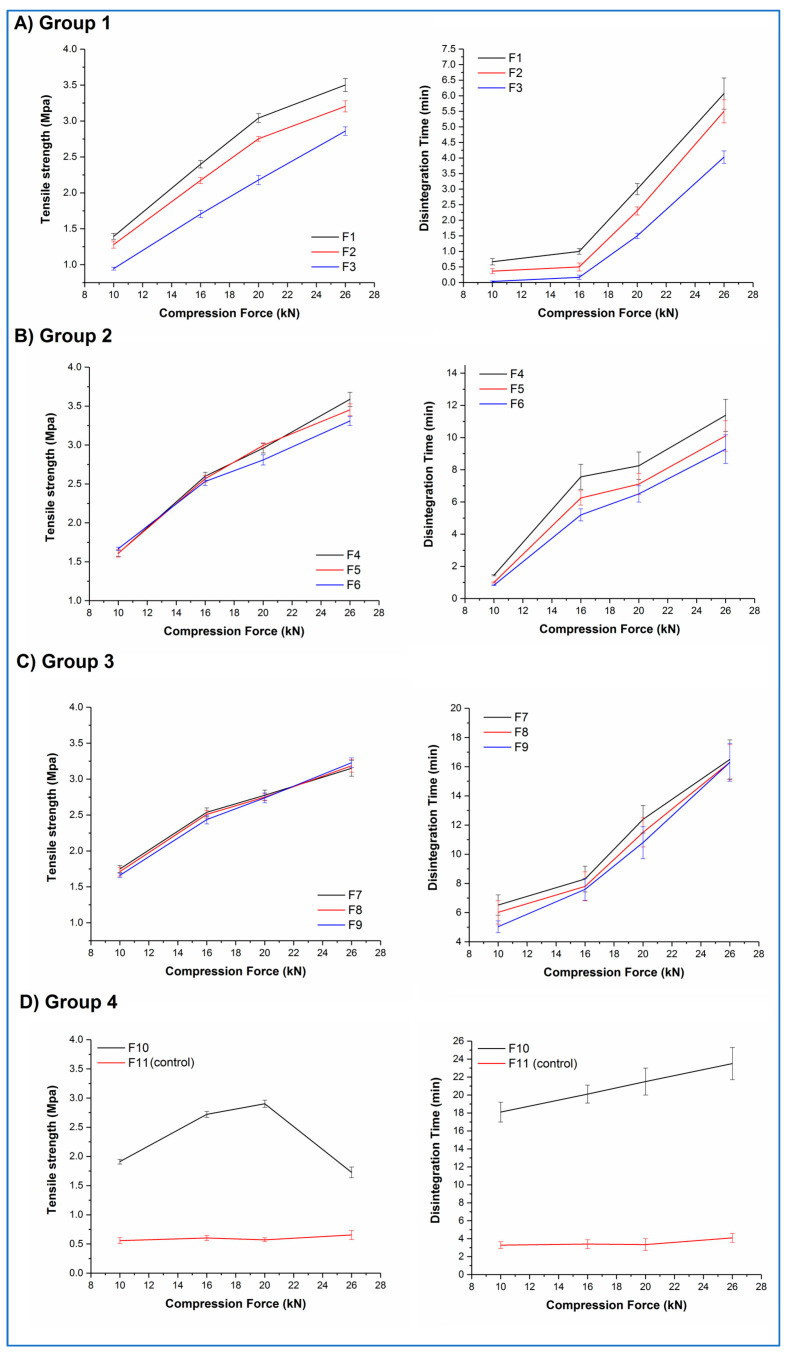
Tensile strength and disintegration time profiles as a function of compression force applied for tablets belonging to (**A**) Group 1, (**B**) Group 2, (**C**) Group 3, and (**D**) Group 4.

**Figure 10 pharmaceutics-15-00802-f010:**
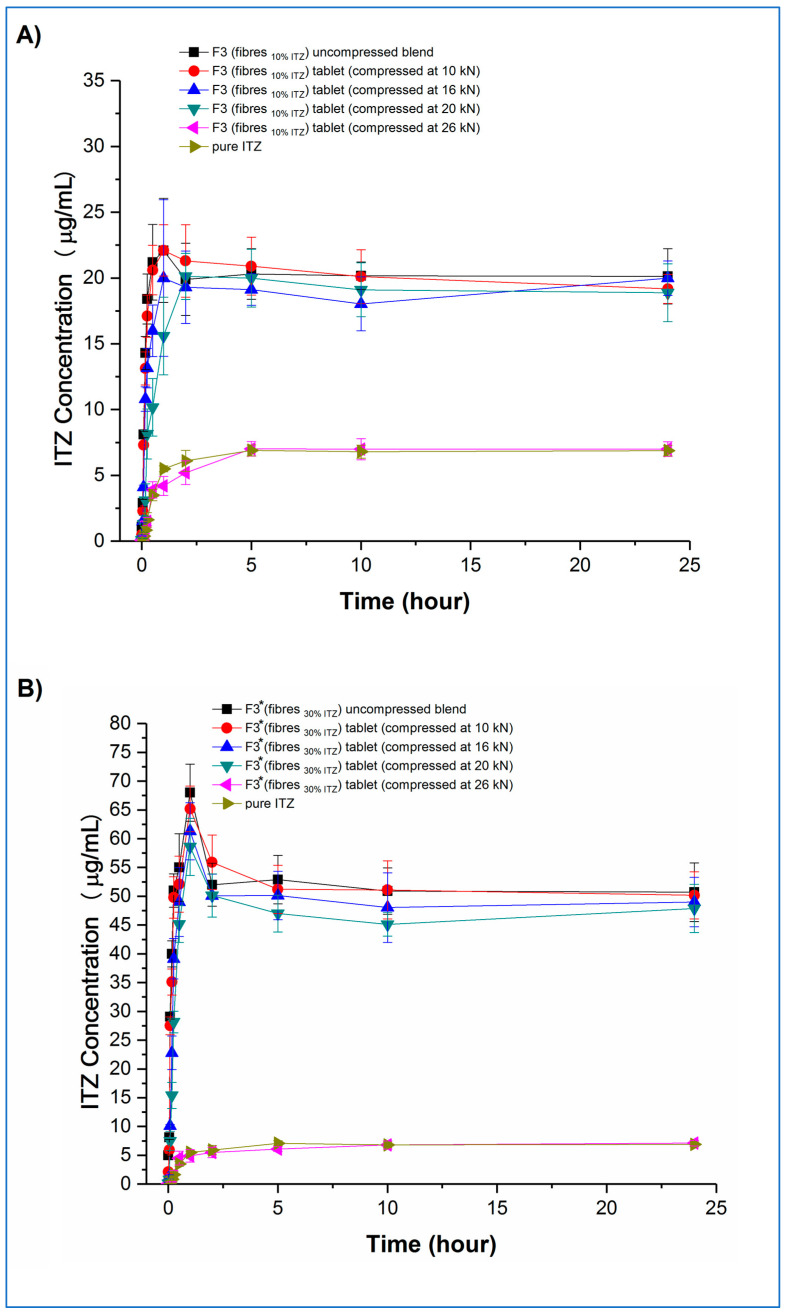
Dissolution–supersaturation profiles obtained under non-sink conditions for tablets, the corresponding uncompressed blends, and pure crystalline itraconazole. (**A**) F3 tablets, and (**B**) F3* tablets. (ITZ = itraconazole).

**Figure 11 pharmaceutics-15-00802-f011:**
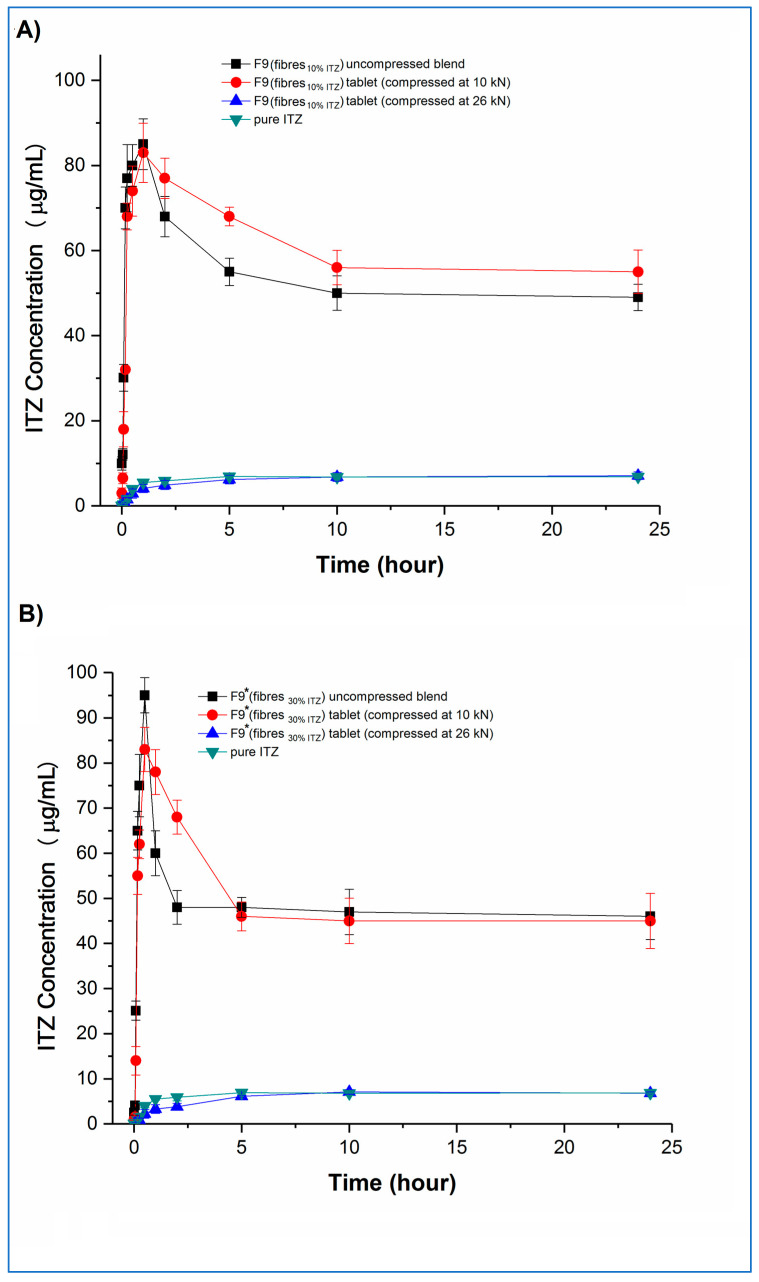
Dissolution–supersaturation profiles obtained under non-sink conditions for tablets, the corresponding uncompressed blends, and pure crystalline itraconazole. (**A**) F9 tablets, and (**B**) F9* tablets. (ITZ = itraconazole).

**Figure 12 pharmaceutics-15-00802-f012:**
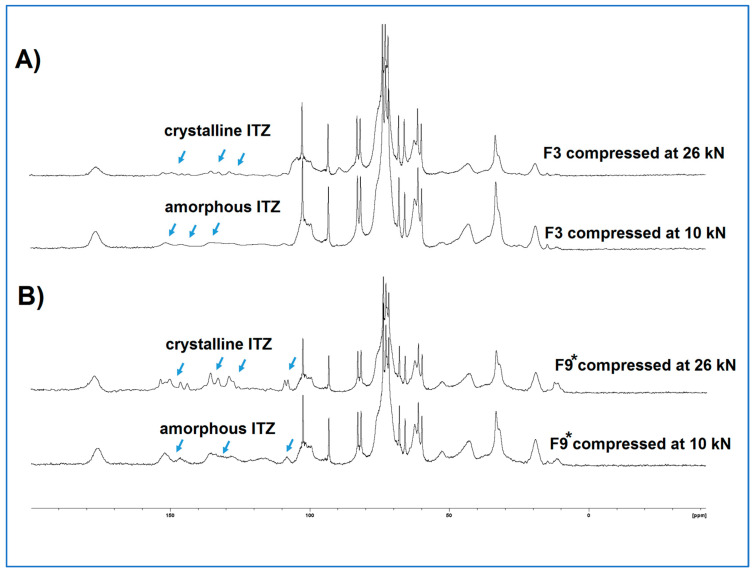
Comparison of ^13^C CP/MAS NMR spectra of tablets compressed at 10 kN and 26 kN for (**A**) F3 tablets (with the lowest itraconazole content) and (**B**) F9* tablets (with the highest itraconazole content). (ITZ = itraconazole).

**Table 1 pharmaceutics-15-00802-t001:** Tablet compositions for the 11 different batches (F1-F11) based on the 10% itraconazole-loaded microfibres (Fibres_10%ITZ_). Values are rounded to one decimal place, so they may not sum to exactly 100%.

Ingredient	Composition (% *w*/*w*)										
	Group 1			Group 2			Group 3			Group 4	
	F1	F2	F3	F4	F5	F6	F7	F8	F9	F10	F11
StarTab^®^	27.8	41.7	50.1	21.1	31.7	38.1	14.5	21.7	26.1	-	-
Avicel PH102^®^	27.8	20.9	16.7	21.1	15.9	12.7	14.5	10.8	8.7	-	-
Compressol SM^®^	27.8	20.9	16.7	21.1	15.9	12.7	14.5	10.8	8.7	-	-
Kollidon CL-F^®^	5.0	5.0	5.0	5.0	5.0	5.0	5.0	5.0	5.0	5.0	5.0
Glyceryl dibehenate	1.5	1.5	1.5	1.5	1.5	1.5	1.5	1.5	1.5	1.5	1.5
Fibres_10%ITZ_	10.0	10.0	10.0	30.0	30.0	30.0	50.0	50.0	50.0	93.5	-
Raw itraconazole	-	-	-	-	-	-	-	-	-	-	9.3
Raw sucrose	-	-	-	-	-	-	-	-	-	-	84.2

**Table 2 pharmaceutics-15-00802-t002:** Average microfibre diameter of freshly prepared samples and particle size (short and long diameter lengths) of the corresponding aged samples after 24 h storage (for microfibres containing 0 and 10% *w*/*w* itraconazole) and 30 days storage (for microfibres containing 20, 30, 50, and 100% *w*/*w* itraconazole) at 25 °C/75% RH.

Itraconazole Loading (% *w*/*w*)	Freshly Prepared Sample	Moisture-Treated Sample	
	Fibre diameter (mean ± SD) (µm)	Long axis diameter (mean ± SD) (µm)	Short axis diameter (mean ± SD) (µm)
0	9.77 ± 3.10	426.29 ± 88.67	301.27 ± 76.11
10 ^a^	6.23 ± 3.88	67.33 ± 29.23	27.48 ± 6.18
20	7.12 ± 2.45	80.13 ± 31.65	26.52 ± 6.78
30	7.49 ± 3.12	83.44 ± 37.23	30.18 ± 6.67
50	6.67 ± 3.89	242.87 ± 45.19	21.52 ± 9.36
100	3.22 ± 2.54	No fibre collapse observed. Fibre diameter (mean ± SD) = 3.13 ± 2.50 µm	

^a^ The data for the pure sucrose and the 10% *w*/*w* itraconazole-loaded samples are reported in [4].

**Table 3 pharmaceutics-15-00802-t003:** Tablet characterisation data for tablet batches F3 and F9 [containing 10% *w*/*w* itraconazole-loaded microfibres (fibres_10%ITZ_)] and F3* and F9* [containing 30% *w*/*w* itraconazole-loaded microfibres (fibres_30%ITZ_)]. (ITZ = itraconazole).

Formulation Code	Compression Force Applied (kN)	Disintegration Time (Seconds or Minutes)	Tensile Strength (MPa)	Weight (mg)	ITZ Content (% of Theoretical)
F3	10	2 ± 1 s	0.60 ± 0.03	299.5 ± 1.1	98.6 ± 1.5
(contains	16	10 ± 2 s	1.09 ± 0.04	301.2 ± 0.9	100.6 ± 1.8
fibres_10%ITZ_)	20	92 ± 11 s	1.35 ± 0.03	298.9 ± 1.2	98.9 ± 2.0
	26	241 ± 24 s	1.71 ± 0.06	300.7 ± 0.7	101.1 ± 1.9
F3*	10	5 ± 2 s	0.66 ± 0.02	299.8 ± 0.9	98.8 ± 0.7
(contains	16	13 ± 2 s	1.13 ± 0.03	300.1 ± 0.8	99.6 ± 1.1
fibres_30%ITZ_)	20	126 ± 21 s	1.28 ± 0.05	299.5 ± 0.8	97.9 ± 1.9
	26	307 ± 39 s	1.77 ± 0.07	300.6 ± 0.9	100.2 ± 2.1
F9	10	5.0 ± 0.4 min	1.06 ± 0.02	298.6 ± 1.3	100.9 ± 2.5
(contains fibres_10%ITZ_)	26	16.3 ± 1.3 min	1.84 ± 0.03	301.2 ± 0.8	101.5 ± 1.9
F9*	10	5.5 ± 0.7 min	1.09 ± 0.04	299.7 ± 0.7	100.9 ± 2.1
(contains fibres_30%ITZ_)	26	16.5 ± 0.9 min	1.73 ± 0.06	298.8 ± 1.4	102.3 ± 1.8

## Data Availability

The data presented in this study are contained within this article.

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
