# Peer review of "Development of Robust Tablet Formulations with Enhanced Drug Dissolution Profiles from Centrifugally-Spun Micro-Fibrous Solid Dispersions of Itraconazole, a BCS Class II Drug"

_pharmaceutics, 2023, doi:10.3390/pharmaceutics15030802_

Round 1

Reviewer 1 Report

Excellent work

Author Response

We would like to thank the reviewer for taking the time to review the paper and for the kind comment.

No changes required.

Reviewer 2 Report

The presented manuscript deals with the problem of enhancing the dissolution profiles of drugs using sucrose microfibers based on the example of itraconazole, a BCS Class II drug. The studied topic is very important and the fact that the Authors try to develop an actual pharmaceutical formulation is particularly valuable and interesting for the scientific community. Therefore, the paper fits very well in the scope of Pharmaceutics.

The introduction describes the current state of knowledge regarding the studied topic, presents some very interesting references, and addresses the previous results obtained by the Authors. While the introduction is rather long, in my opinion, it should be retained because it gives a very good insight into the presented topic.

The methodology is well described, with references to previous works, and allows both for the replication of the results and using it for other systems. The number of used techniques shows that the Authors have conducted a very thorough analysis of the studied systems enabling their full characteristics.

The obtained results are credible and very thoroughly discussed. All the systems are adequately described and the observed phenomena are clearly explained, which gives an insight into the mechanisms responsible for the observed results. All of my initial concerns were quickly dispelled with the unfolding discussion of the results.

Some very important findings were made during the studies. The fact that the samples of itraconazole-loaded sucrose microfibers retain their dissolution and supersaturation advantage after elongated humidity treatment is rather surprising and made it possible to utilize such aged samples for tablet formulation. The tablet composition and preparation conditions were optimized and led to obtaining an effective pharmaceutical formulation with good dissolution behavior, which is particularly valuable.

Overall, the presented manuscript has a very high scientific value. It shows a method of increasing dissolution that can be easily applied to other systems, explains the observed phenomena, and is an excellent starting point for further studies. Its clarity and used language allow for easy reading and understanding of the presented concepts.

Therefore, with pleasure, I can recommend the acceptance of the manuscript in Pharmaceutics in its current form.

Author Response

We would like to thank the reviewer for taking the time to review the paper and for the kind comments.

No changes required.

Reviewer 3 Report

The current research focusing on the compressibility of amorphous microfibers into tablet formulation is interesting and novel. The entire content is well-written. The manuscript needs to be cross-checked for minor typos and grammatical errors. Authors are requested to address the following comments before considering the manuscript for publication.

1.     Please make a note of the drug's thermal properties.

2.     Line 181: Please correct the following “10 °C/minute to 3 °C”

3.     How was an inert atmosphere maintained in the DSC study? Please capture the methodologies in detail.

4.     Section 2.1.2: How were the conditions maintained?  

5.     Section 2.1.3: Please describe the methodologies in detail.

6.     Please cross-check the composition of magnesium stearate.

7.     What temperature was employed for the preparation of 100% pure itraconazole fibers?

8.     How long does it take to prepare the fibers for each batch? If the output rate is similar to the electrospinning process, do authors think it can be a commercially viable process for future use?

Author Response

Please see attached PDF document.
